# A Multiplex Analysis of Potentially Toxic Cyanobacteria in Lake Winnipeg during the 2013 Bloom Season

**DOI:** 10.3390/toxins11100587

**Published:** 2019-10-11

**Authors:** Katelyn M. McKindles, Paul V. Zimba, Alexander S. Chiu, Susan B. Watson, Danielle B. Gutierrez, Judy Westrick, Hedy Kling, Timothy W. Davis

**Affiliations:** 1Department of Biology, Bowling Green State University, Bowling Green, OH 43403, USA; kmckind@bgsu.edu; 2Center for Coastal Studies, Texas A&M University—Corpus Christi, 6300 Ocean Dr., Corpus Christi, TX 78412, USA; Paul.Zimba@tamucc.edu; 3Base Pair Biotechnologies, 8619 Broadway St, Suite 100, Pearland, TX 77584, USA; alex.chiu@basepairbio.com; 4Department of Biology, University of Waterloo, Waterloo, ON N2L 3G1, Canada; jkswatson@shaw.ca; 5Mass Spectrometry Research Center, Department. of Biochemistry, Vanderbilt University, 9160 Medical University Bldg III, 465 21 Ave South, Nashville, TN 37240, USA; d.gutierrez@vanderbilt.edu; 6Department of Chemistry, Wayne State University, Detroit, MI 48202, USA; judy.westrick@wayne.edu; 7Algal Taxonomy and Ecology Inc., PO Box 761, Stony Mountain, MB ROC 3A0, Canada; hedy.kling8@gmail.com

**Keywords:** cyanobacteria, cylindrospermopsin, microcystin, multiplex qPCR, Lake Winnipeg, saxitoxin

## Abstract

Lake Winnipeg (Manitoba, Canada), the world’s 12th largest lake by area, is host to yearly cyanobacterial harmful algal blooms (cHABs) dominated by *Aphanizomenon* and *Dolichospermum*. cHABs in Lake Winnipeg are primarily a result of eutrophication but may be exacerbated by the recent introduction of dreissenid mussels. Through multiple methods to monitor the potential for toxin production in Lake Winnipeg in conjunction with environmental measures, this study defined the baseline composition of a Lake Winnipeg cHAB to measure potential changes because of dreissenid colonization. Surface water samples were collected in 2013 from 23 sites during summer and from 18 sites in fall. Genetic data and mass spectrometry cyanotoxin profiles identified microcystins (MC) as the most abundant cyanotoxin across all stations, with MC concentrations highest in the north basin. In the fall, *mcyA* genes were sequenced to determine which species had the potential to produce MCs, and 12 of the 18 sites were a mix of both *Planktothrix* and *Microcystis*. Current blooms in Lake Winnipeg produce low levels of MCs, but the capacity to produce cyanotoxins is widespread across both basins. If dreissenid mussels continue to colonize Lake Winnipeg, a shift in physicochemical properties of the lake because of faster water column clearance rates may yield more toxic blooms potentially dominated by microcystin producers.

## 1. Introduction

Lake Winnipeg (Manitoba, Canada) is the world’s 10th largest freshwater and the 12th largest lake overall by surface area (24,514 km^2^). The lake is separated into three distinct regions: a shallow south basin (SB; Zm = 9 m), with high turbidity from runoff from the Red River and sediment resuspension [1], the Narrows, where restricted flow results in sediment deposition, and a larger and deeper north basin (NB) with clearer water (>80% total surface area, Zm = 13.3 m). Of the two primary tributaries, the Red and Winnipeg Rivers, the Red River flows through extensive agricultural lands before entering the SB of the lake [2,3]. The watershed spans nearly 1 million km^2^ and encompasses four provinces and two U.S. states [4]. The lake generates significant revenue including major contributions to Manitoba’s $30 million/year fishing, $100 million/year tourism, and >$350 million/year hydroelectric industries [5,6].

Over the past several decades, this major freshwater resource has shown an alarming decline in water quality as highlighted by historical data and core samples that have shown a distinct increase in phosphorus, carbon, and chlorophyll-*a* (chl-a) since the late 1950s [7,8]. The gradual rise in nutrient levels in the lake because of basin development over the past century was exacerbated by the construction of hydroelectric dams on some of the major inflows (e.g., Winnipeg and Saskatchewan Rivers) and at the northern outlet (Nelson River), creating a major step change in hydrology, flushing, and nutrient retention [9]. This has been accompanied by climate change and increases in precipitation and flooding since the 1990s. Total phosphorus (P) concentrations have doubled between 1990 and 2000, and currently the lake is considered hypereutrophic [10] and was declared the “threatened lake of the year” by the Global Nature Fund [11].

Satellite imagery and in-lake biomass measures of Lake Winnipeg indicate that the frequency and magnitude of severe summer–fall cyanobacterial harmful algal blooms (cHABs) has increased [8,12]; however, they differ significantly year to year in associated toxin concentration and species composition [13]. Microcystins (MCs) are the most ubiquitous group of cyanotoxins and are found on every continent including Antarctica [14,15,16]. MCs have been detected throughout inland waters in North America [17,18,19,20], and low levels have been reported by several studies from Lake Winnipeg [21,22,23]. MCs can be produced by some species belonging to a variety of cyanobacterial genera including *Microcystis*, *Dolichospermum, Planktothrix, Nostoc, Hapalosiphon, Pseudanabaena, Tolypothrix, Oscillatoria, Phormidium, Rivularia*, and *Anabaenopsis* [24,25,26,27]. Some of these taxa are present in the Lake Winnipeg cHABs, along with the epiphytic MC producer, *Pseudanabaena rutilis-viridis,* although the bloom assemblages are largely dominated by diazotrophic species (notably *Aphanizomenon flos-aquae* complex, *Dolichospermum flos-aquae* and *Dolichospermum lemmermannii*) [8,13]. Sampling to date has only detected comparatively low concentrations of MCs and despite the presence of cyanobacteria that can potentially produce other toxins (e.g., saxitoxins, anatoxins, and cylindrospermopsins [28]), these metabolites have not been detected previously in Lake Winnipeg [21,22,23]. Lake Winnipeg is threatened further by the recent appearance of invasive dreissenid mussels [29], which may result in a major shift in nutrient, light regimes, bloom composition, and toxin concentration [30]. Algal blooms in Lake Winnipeg are not new and have been documented as far back as 1934 [8,31], but the current situation represents a potential regime shift and major threat to this important freshwater’s biodiversity.

Toxic and non-toxic (i.e., not genetically capable of producing known cyanotoxins) strains of the same species often co-occur in the same bloom but are indistinguishable using traditional identification and enumeration techniques [32]. Therefore, the development of molecular techniques has become an invaluable tool, allowing a far greater capacity to resolve questions about taxa and toxicity. In the early 2000s many molecular targets were used to detect toxin-producing genes in water samples with a theoretical detection limit of a single gene copy, even when no toxin was detectable at the time of sampling [33,34,35,36]. Further development of these methods has allowed the detection and quantification of genes for multiple toxins from a sample using a single assay [36]. The multiplex qPCR method described by Al-Tebrineh et al. [36] can simultaneously identify certain genes within the gene clusters *mcy*, *cyr*, and *sxt* responsible for the production of MCs, cylindrospermopsins (CYNs), and saxitoxins (STXs), respectively. This approach allows a more detailed assessment of the genetic potential for toxin production than more traditional single-gene assays, and coupled with other measures, provides insight into the presence and expression of toxin-producing genes within a planktonic community.

The primary objective of this study was to establish a baseline assessment of the pre-dreissenid genetic potential for toxin-production and associated taxa in Lake Winnipeg. This was addressed using a multiplex qPCR method to evaluate the spatial-temporal distribution of *mcy*, *cyr*, and *sxt* genes in summer and fall water samples collected from sites across the SB and NB during research cruises in 2013 (Figure 1). Samples were also analyzed for toxin concentrations, and these data were combined with measures of key environmental parameters such as dissolved and particulate nutrient levels and temperature. Fluoroprobe^©^ data was also collected, to evaluate the biomass and proportions of major phytoplanktonic groups present in the water column. With this approach, we can simultaneously investigate the genes controlling the production of the three major toxin groups, MCs, CYNs, and STXs in Lake Winnipeg, the dominating organisms making up the cHABs, and their potential link with environmental drivers.

## 2. Results

### 2.1. Nutrients

Nitrogen (N) was measured at all sites as NH_4_^+^, NO_2_^−^ - NO_3_^−^, and TKN-F. Additionally, particulate P (PP), SRP, and TP concentrations were measured at 17 of the 23 summer sites. The highest concentrations of all N fractions and TDP were detected in the SB during the summer sampling (NH_4_^+^ = 53 N μg/L, NO_2_^−^ - NO_3_^−^ = 509 N μg/L, and TKN-F = 749 N μg/L), while the maxima for PP and TP were measured at site W3 on the western shore of the NB (PP = 88 μg/L, TP = 95 μg/L; Table A1). In the summer, nutrient fraction concentrations for TDP and TDN (NO_2_^−^-NO_3_^−^, + TKN-F) were highest in the south of the lake and declined northward. This gradient corresponds to positive correlations across the lake with 0.71 < *r* < 0.80; *p* < 0.002).

During the fall, maximum concentrations of NO_2_^−^ - NO_3_^−^, TDN, TKN-F, TDP, and TP occurred at the south end of the lake (site 3B; NO_2_^−^ - NO_3_^−^ = 365 N μg/L, TDN = 997 N μg/L, TKN-F = 632 N μg/L, TDP = 205 μg/L, and TP = 240 μg/L) (Table A2). The maximum concentrations of NH_4_^+^ and PP were measured in the north east of the lake at sites 21 and 2-mile inlet respectively. Concentrations of TDP were again highly correlated with NO_2_^−^ - NO_3_^−^ and TKN-F, with r values of 0.93 (*p* < 0.0000001), 0.94 (*p* < 0.0000001), and 0.81 (*p* < 0.00005) respectively. 

### 2.2. Fluoroprobe Analyses

Fluoroprobe data was collected during the summer cruise only (Figure 2). These data showed a maximum value of 34.5 µg/L chl-a, contributed largely by cyanobacteria (86%), recorded at site W3 on the western shore of the NB, where the highest levels of PP and TP were also recorded. Diatom/dinoflagellates were found in most locations throughout the lake at low abundance with a maximum concentration of 3.8 µg/L chl-a at site W2 in the northeast. Green algae and cryptophytes were detectable at only five and three out of 22 sampling sites respectively, with maxima of 2.0 µg/L chl-a for green algae at the mouth of the Red River in the south (site 3B) and 4.8 µg/L chl-a for cryptophytes in the northeast at site W3. 

### 2.3. Community Composition

Samples analyzed for community composition include eight summer sites and six fall sites (Figure 3). With the exception of three summer sites (13b, 23es, 67NS), the majority of the biomass at each sampled site in both seasons was from Cyanobacteria (Table 1), in particular from *Aphanizomenon flos aquae* complex and *Dolichospermum* spp. Also of note was that *Planktothrix agardhii* was a dominant non-diazotroph species at fall site 31 and 28. Diatoms were the next most abundant organisms in Lake Winnipeg, including dominance at summer site 67NS. The least abundant phylum were the Dinoflagellates, which were often absent from the summer samples entirely. 

### 2.4. Potentially Toxic Gene Abundance and Toxin Concentration

Multiplex qPCR analyses of the 23 sites sampled during summer revealed that the *cyr*A, *sxtA*, and *mcyE* genes were detected at 6, 2, and 17 sites, respectively (Table 2; Figure 4, Figure 5 and Figure 6). All sites that tested positive for *cyrA* and *sxtA* genes also had *mcyE* genes, with the two *sxtA* sites positive for all three toxin producing genes. The highest recorded gene densities were 113 ± 84, 16 ± 15, and 4.41 × 10^4^ ± 1.89 × 10^4^ genes/mL for *cyrA*, *sxtA,* and *mcyE,* respectively, all of which were detected on the eastern side of the NB (Figure 4a, Figure 5a, and Figure 6). Nine of the 23 sites were positive for MC using ELISA (Figure 4b); most of these were located in the NB, with one (site 3B) at the mouth of the Red River in the southernmost part of the lake. All of these sites were positive for *mcyE* genes and eight of the nine also returned positive readings for MC via mass spectrometry multiple reaction monitoring (MS-MRM) analysis (Figure 4), the one site that returned a negative result being 3B. The maximum MC concentration using ELISA was 9.2 µg/L at site W3, on the eastern side of the NB, the same site having the highest MC concentration by MS-MRM (14.2 µg/L) and *mcyE* gene abundance. MCs were detected using MS-MRM at 14 sites located predominantly on the eastern side of the NB (Figure 4c). The majority (86%) of the 14 positive MS-MRM sites were also positive for the presence of *mcy* genes. 

Just under half (10) of the summer samples were also positive for CYN, but *cyr* genes were detected at only 60% of these sites (Figure 5). In contrast to microcystins, which were most abundant in the NB, the maximum CYN concentration (1.37 µg/L) was measured at site 2 in the southern end of the lake. The second highest concentration of 0.53 µg/L occurred at site 22 (in the north eastern region of the lake), where the highest copy number of *cyr* gene was also detected (Figure 5). No STX was detected by MS-MRM. A comparison of the MC/*mcy* data generated via qPCR, MS-MRM, and ELISA showed good correspondence among the three methods (Table 3).

During the fall, the 18 sites sampled included three *cyrA*, no *sxtA,* and 18 *mcyE* positive locations (Table 4; Figure 7a and Figure 8a). As with the summer samples, sites that tested positive for *cyrA* were positive for *mcyE* genes. The maxima concentrations for the *cyrA* and *mcyE* were 11 ± 4 and 6410 ± 2090 genes/mL respectively (Table 4) which were an order of magnitude lower than the peak levels observed in the summer (Table 2). The peak abundances of *mcyE* and *cyr*A were observed in the NB in both seasons; however, in the summer they were concentrated on the east coast of the NB, whereas in the fall they were concentrated in the north-western corner of the lake (Figure 7a and Figure 8a). Furthermore, while *mcyE* was the only gene detected at one out of the seven SB sampling sites in the summer, it was detected at all sampling sites throughout the lake in the fall, in generally lower abundances. There were too few positive sites to make any meaningful comment on the distribution of *cyrA* in fall. Compared to the summer, toxin levels were generally lower in fall (often close to the limit of detection) and showed different spatial patterns than in the summer. Of the 18 sites tested using MS-MRM, only five and two sites tested positive for MC (Figure 7c) and CYN (Figure 8b) respectively, while none were positive for STX. The peak levels of MC and CYN detected in fall were very low (0. 154 and 0.007 ug/L respectively), approximately two orders of magnitude lower than the summer maxima. In the summer survey, MC positive sites were largely located in the NE, while in the fall, MC was detected at the very north end of the lake and also in the narrows (Figure 4 and Figure 7). While CYN was detected at over 40% of sampling sites in the summer (80% of these in the NB), it was only detected at two sites (11%) in the fall, one in the narrows and one in the SB (Figure 8).

### 2.5. mcyA Phylogenic Analysis

All the fall sites and two summer sites were chosen for the *mcyA* sequence analysis to investigate the sample for potential MC-producing species. The sequences from each site were compared to reference sequences from known MC-producers, and a representative sequence from each species was pulled to generate a Maximum-likelihood phylogenetic tree (Figure 9). A majority of the sites (12 of 20) contained a mixture of both *Planktothrix* and *Microcystis* sequences. Four of these mixed sites were dominated by *Planktothrix* (NB sites 28 and 41S, SB sites W11 and 5) and eight were dominated by *Microcystis* (NB sites 2-mile inlet, 34S, W2, 23S, 39, 43S, the Narrows site 64, and the SB site 60C) (bolded sites, Figure 9). Of the single species sites, six were *Microcystis* only (NB sites 22, 21, the narrows site 56, and the SB sites W10, 58Sn and 3B) and two were *Planktothrix* only (NB site 31 and the narrows site W8). There were no observed trends in the location of these sites and the presence of one MC producing species vs. the other. 

### 2.6. Correlations

For the summer data, strong positive correlations were observed among *mcyE* gene abundances and both MS-MRM and ELISA MC data, with Pearson’s correlation coefficients of 1.0 (*p* < 0.001), 0.99 (*p* < 0.001), and 0.99 (*p* < 0.001) for qPCR vs. MS-MRM, qPCR vs. ELISA, and MS-MRM vs. ELISA respectively (Table 3). No significant correlations were observed between the qPCR and MS-MRM data for CYN/*cyrA* and STX/*sxtA* which had much smaller sample sizes. There were also strong positive correlations between PP and concentrations of *mcyE* and MC with r values of 0.90 (*p* < 0.001), 0.90 (*p* < 0.001), and 0.96 (*p* < 0.001) for PP vs. qPCR, MS-MRM, and ELISA respectively. A strong positive correlation between fluoroprobe measures of cyanobacterial abundance and both MC and *mcyE* data, (r values of 0.84 (*p* < 0.001), 0.86 (*p* < 0.001) and 0.90 (*p* < 0.01) for cyanobacteria vs. qPCR, MS-MRM, and ELISA respectively).

During the fall, there were fewer significant correlations among the different measures, reflecting the generally lower levels of toxins and toxin gene copies across the lake. No significant correlations were found between the nutrient concentrations and toxin and/or gene abundances in the fall. Nevertheless, significant positive correlations were observed between *mcyE* and MS-MRM (*r* = 0.63, *p* < 0.01), and *mcyE* and ELISA (*r* = 0.78, *p* < 0.05), although the relationship between MS-MRM and ELISA was not statistically significant (*r* = 0.50, *p* = 0.09) (Table 3).

## 3. Discussion

While we understand that these data are from 2013, we feel it is important to report these values as a historical data set in which future studies can refer back to. The current study aimed to set baseline measurements for seasonal bloom toxin concentration as well as potential cyanotoxin producers in Lake Winnipeg, which marks an important first step which could be used to compare to the conditions in the lake after dreissenid colonization. Additionally, the data presents information regarding the key players in Lake Winnipeg blooms, which can be used for further research behind understanding the drivers of bloom dominance. This work is one of only a few studies to combine multiplex qPCR and toxin analysis to evaluate the bloom assemblages. Additionally, our work both demonstrated the potential value of this multiplex approach and provided important new insight into the capacity and expression of a range of cyanotoxins by Lake Winnipeg cHABs. Consistent with earlier studies on this lake, our data showed that the current bloom assemblages produce generally low levels of MC toxins, largely during the summer in the NE regions, likely reflecting both the higher transparency of the NB compared to the more turbid southern basin (a factor that has been linked to increased microcystin production) [37,38,39], and wind-driven movement of surface material toward the eastern shoreline [15,40]. 

In addition, our multiplex qPCR data show that the production capacity for MC and other cyanotoxins is widespread across both basins, demonstrating a potential for more toxic blooms in this lake that future changes in food web or climate regimes could facilitate. Lake Winnipeg summer blooms in 2013 were largely dominated by cyanobacteria (Figure 2 and Figure 3) as reported from previous years [8], and this study demonstrates that at least two cyanotoxin groups (MCs and CYNs) were detected via ELISA or MC-MRM during the 2013 bloom and the genetic capability to produce MCY, CYN, and STX were also observed. While this is concerning, it is not unique as blooms that have the genetic capability to produce multiple toxins have been found elsewhere [41]. Of these, in Lake Winnipeg, MC was the most commonly detected and has been previously reported [21,22,23]. Additionally, the detection of CYN/*cyrA* demonstrates that potential CYN-producers are expressing these toxins [28], while the detection of *sxtA* genes at two sites highlights the potential future threat of this third group of cyanotoxins known to cause severe health problems in many locations worldwide [42]. While we did not have the opportunity to analyze samples for the genetic potential to produce yet another cyanotoxin, anatoxin-a, as it was not included in the multiplex qPCR kit; this toxin was not identified by HPLC-MS indicating that the current risk is very low but investigating the genetic potential to produce anatoxin-a remains an important knowledge gap for future studies. This study also was unable to address the potential presence of β-*N*-methylamino-*L*-alanine (BMAA), a neurotoxin produced by some cyanobacterial species. BMAA has previously been detected in Lake Winnipeg [43,44], ensuring that future work should include this toxin and its isomers as well as the ones analyzed in this work.

Further analysis of the potential MC producers in the lake (which make up only a small fraction of the total cyanobacterial population) indicated the presence of co-existing *Microcystis aeruginosa* and *Planktothrix agardhii* (Figure 9). These two species are expected to occupy different niches, as *Planktothrix agardhii* is considered to be low-light adapted and tends to dominate in more turbid regions [45], while *Microcystis aeruginosa* is usually found in locations with high nutrients, light and temperatures and low turbulence [46,47]. It is likely that this co-occurrence at many of the sites reflected small abundances of *Planktothrix* and *Microcystis* present at each site relative to the much more dominant cyanobacteria. Manual cell counts were unable to find discernable populations of *Microcystis* and *Planktothrix* at certain sites in the fall (Table 1), but because the sequencing work was only looking for the potential MC producing organisms, it is probable that the sequences from each site were derived from only a few number of cells. The State of Lake Winnipeg report [48] addresses the fact that while *Microcystis* blooms can occur in Lake Winnipeg, they are not consistent. Future work in the area will include more genetic analysis to determine if one of these potential toxin producers starts to dominate over the other in sites where we found genetic markers for the presence of both species.

As expected, the lake showed a general nutrient gradient from the south to the north, consistent with the majority of the nutrient load to Lake Winnipeg entering from the major tributaries in the south and the general south-north flow [2,3]. One notable exception was seen with particulate P, which had maxima concentrations in the NB in both seasons, suggesting that the toxic bloom is dependent on P supply, as indicated by the strong positive correlation between PP and measures of MC and *mcy* shown previously [49]. While the nutrient analyses in 2013 did not include total N, the dominance of diazotrophs (Table 1) would suggest that N limitation is a common feature of this lake at its present trophic state. Based on the environmental data we have presented, Lake Winnipeg blooms are strongly affected by N and P concentrations, light intensity through water turbidity, and prevailing wind direction. 

The strong positive correlation between qPCR, MS-MRM, and ELISA for MC/*mcy* quantification suggests all three methods provided consistent results. While the ELISA and MS-MRM methods were correlated, there were minor differences in the detection of MC which can likely be attributed to the fact that the ELISA method has lower sensitivity than HPLC methods. At the low concentrations detected here (Table 2 and Table 4), more will be visible using MRM procedures than the ELISA if the congener standard is available. The recently developed qPCR method further tested here offers the potential for a valuable early warning system to detect and quantify the presence and expression of toxin genes and can be used for risk assessment in the summer. In the fall, qPCR may still provide a weaker predictor of MC concentration, but overall indicates the need to continue to monitor for actual toxin concentrations later in the bloom. 

The current blooms in Lake Winnipeg are similar to the Lake Erie blooms of the 1950s–1970s. Excessive inputs of nutrients and suspended material during the first half of the 1900s led to significant cHABs across Lake Erie which were primarily dominated by diazotrophic, turbidity tolerant species of *Aphanizomenon* and *Anabaena* (which is the same genus as Lake Winnipeg *Dolichospermum*). Following the restoration of this and the other Laurentian Great Lakes under the US-Canada Great Lakes Water Quality Agreement (1968; amended 1972), these blooms dissipated. As water quality improved, binational monitoring efforts were decreased; however, during this time, invasive dreissenid mussels were introduced into the Great Lakes and became established in Lake Erie in the 1990s. In parallel, non-point source nutrient inputs into Lake Erie also increased leading to the annual blooms dominated by toxic *Microcystis* [50,51]. It is hypothesized that the filter-feeding dreissenid mussels contributed significantly to dramatic shifts in the food web composition and/or water column clarity, and augmented levels of bioavailable dissolved nutrients, facilitating the dominance of *Microcystis* over *Aphanizomenon* or *Dolichospermum* [50,52]. However, because of lack of monitoring during the establishment of the dreissenids, this hypothesis cannot be tested. The recent appearance of dreissenids in LW provides an opportunity to test this hypothesis, as conditions in this lake (light and nutrient regimes, and cHAB composition) are similar to those seen during the pre-dreissenid period of Lake Erie. Hence, if dreissenids colonize Lake Winnipeg successfully, continued alteration of the plankton community by preferential filtration of non-colonial forms [53,54] may aid in shifting bloom dominance toward *Microcystis*, potentially leading to more toxic blooms and shifts in lower trophic level dynamics, significantly impacting the ecology of the lake and socioeconomics of the region. The potential for an analogous scenario to Lake Erie in Lake Winnipeg is sobering reality, given the similarities of these two lakes.

## 4. Conclusions

This work has set a baseline to evaluate the effects of a potential regime shift in the phytoplankton communities in Lake Winnipeg. The taxa and genes for toxin production exist at low levels in Lake Winnipeg and given the response of Lake Erie to a similar invasion of dressinids, may result in a shift toward more toxic blooms. These results also demonstrate the use of multiplex q-PCR pre-bloom to access and monitor water bodies for toxin-forming potential. The ability of this approach to detect toxin genes and the associated toxins produced was validated using multiple analytical method. This research also confirmed the widespread presence of both the genetic potential and expression of two important classes of cyanotoxins, cylindrospermopsin and saxotoxins, in Lake Winnipeg. Currently, these are present at low levels; however, future changes in climate and foodweb regimes has the potential to sign alter this production, indicating a need for research on the drivers of Lake Winnipeg cHABs post dressinid colonization. The *cyrA* and *sxtA* data highlight the advantage of using a multiplex qPCR assay over more traditional qPCR single target assays for a more comprehensive measure of bloom toxin potential. The qPCR data also suggest that some Lake Winnipeg bloom assemblages contained strains capable of producing multiple toxins or a mixed population of species capable of producing different toxins. The co-occurrence of three different toxin genes at very low concentrations in the lake at individual locations raises the concern that lake users (human and non-human) may, in the future, be subjected to synergistic toxic effects that are potentially stronger than exposure to any individual toxin.

## 5. Materials and Methods 

### 5.1. Water Collection, Filtration, and Microscopy

Water samples were collected from 23 sites in June 2013 (summer) and 18 sites in September 2013 (fall) (Figure 1) as part of several research cruises. Sample locations varied between the seasons based on the collective need of the researchers on the boats. Water was pumped from approximately 1 m beneath the lake surface through an intake located underneath the center of the RV *Namao*. Triplicate samples were filtered (summer used 47 mm diameter 1.2 um pore size GF/C filters Millipore Corporation, Jaffrey, NH, USA; fall used 47 mm diameter 1.2 um pore size Polycarbonate filters Millipore Corporation, Jaffrey, NH, USA) via vacuum filtration (<15 torr) and the volume of water was recorded. The volume of water ranged from 100 to 750 mL per filter, depending on the site. Filters were placed into cryo vials and stored at −80 °C until processed. Two filters were used for molecular analysis and one was used for instrumental toxin analysis as described below. Additional samples were collected and fixed at a 5% final concentration Lugol’s solution for microscopy analysis of biomass and species composition.

Phytoplankton and planktonic protozoans were considered to have the same density as water. A maximum of 2 mL subsamples of the Lugol’s preserved whole water sample are settled in a Utermöhl settling chamber. Samples were allowed to settle for 4 or more hours to allow even the tiniest cells to settle. An inverted microscope was used for sample identification and enumeration using magnifications between 125× and 625×. The small (<20 μm) numerous cells are enumerated on a 250 μm transect at the higher magnification and the larger (>20 μm cells) less numerous cells are enumerated at the lower magnification on ½ of the chamber. At least 200 organisms are enumerated and classified to the lowest taxonomic entity possible in a reasonable time frame. Using standard geometric shapes for the cells and a specific gravity of one biomass conversions are calculated.

Rational for number of organisms counted is based on QAQC studies which have evaluated counting and sampling errors with the Utermöhl inverted microscope method [55,56,57,58,59].

Taxonomic identifications are based on several morphological taxonomic guides and keys listed in Freshwater Algae of North America first and second editions. Taxonomy is still evolving and with the recent multiphasic approach using natural morphology, culturing, cytology, ecology and sequencing and the current microscopic analysis documents the taxa-based features visible at the time of capture. 

### 5.2. Dissolved Nitrogen (N) and Phosphorus (P)

Single water samples from each site were filtered through a membrane filter (0.45 µm 47 mm cellulose acetate; Sartorius Biotech Inc., Goettingen, Germany). Aliquots were stored at 4 °C until processed for total dissolved N, NO_2_^−^, NO_3_^−^ at the University of Alberta (Edmonton, Canada) using standard methods [60]. Soluble reactive P (SRP) was measured aboard the ship with minimal delay (~1–4 h) after samples were collected using the standard ascorbic acid-method described by Stainton et al. [61]. Total phosphorus (TP) and particulate P (PP) were measured at the National Laboratory for Environmental Testing (Burlington, Canada) using standard methods [60]. 

### 5.3. Fluoroprobe

At each sampling site during the summer cruise a fluoroprobe (bbe Moldaenke, Schwentinental, Germany) was lowered to approximately 1 m below the water surface. The probe estimates the abundances of four major groups of phytoplankton, “blue-green” algae (Cyanobacteria), “green” algae (chlorophytes and euglenophytes), “brown” algae (diatoms, chrysophytes and dinoflagellates), and “red” algae (cryptophytes) in the surrounding water based on spectral fluorescence patterns and manufacturer calibrations. No correction was made for background chromophoric dissolved organic material. Data were visualized using Ocean Sneaker’s Tool software (version 2.0.0.51, open source from agris.fao.org).

### 5.4. DNA Extraction, Multiplex qPCR Analysis, and mcyA Sequencing

Duplicate frozen filters were thawed and extracted using the PowerPlant^®^ Pro protocol (MO BIO Laboratories, Carlsbad, CA, USA) with 0.1 mm glass beads for 3 min at 2800 rpm using a BeadBug™ microtube homogenizer (Benchmark Scientific, Edison, NJ, USA). Samples were eluted into a total volume of 100 µL. Absorbance readings of DNA samples were taken at 260 nm and 280 nm using a Take3™ micro volume plate on a Synergy plate reader (BioTek^®^ Instruments, Winooski, VT, USA) to determine the quality and quantity of DNA in each sample. Extracted DNA samples were stored at −20 °C until analysis.

Molecular grade H_2_O (80 µL) was added to each tube of a CyanoDTec™ cyanotoxin detection kit (Diagnostic TECHNOLOGY, Sydney, Australia) and processed following kit directions. The kit comes with fully validated primers and probes for the detection of total cyanobacteria (16S) as well as an internal amplification control to determine if any PCR inhibitors existed in the individual extracts. The kit also included with fully validated primers and probes for the quantification of the *cyrA*, *sxtA*, and *mcyE* genes, which are involved in the gene operons linked to the production of cylindrospermopsins, saxitoxins, and microcystins, respectively. A synthetic standard of known toxin gene copy (Diagnostic TECHNOLOGY, Sydney, Australia) was assayed in serial dilutions spanning five orders of magnitude (1 × 10^6^–1 × 10^2^) to generate a standard curve for each aforementioned target toxin gene. Thermal cycling and fluorescence detection was performed using a Bio-Rad iQ™5 real-time PCR detection system (Bio-Rad, Hercules, CA, USA). Threshold cycle values for each sample were determined using Bio-Rad iQ™5 Optical System Software (version 2.1.97.2001, Bio-Rad, Hercules, CA, USA). Gene copy/mL for each water sample was calculated after normalization for test procedures.

PCR amplifications were then performed on 18 fall and two summer field samples using *mcyA* primers that detect potential microcystin-producing genotypes in *Microcystis*, *Planktothrix,* and *Dolichospermum* [54]. These primers have been used in previous cHAB phylogenetic studies, including work on the cHABs of the Great Lakes [35,45,62,63,64]. PCR conditions were similar to those described in Hisbergues et al. [62]. Briefly, an initial denaturation at 95 °C for 10 min; 40 cycles of 94 °C for 30 s, 59 °C for 30 s, 72 °C for 30 s, and a final extension step at 72 °C for 5 min were performed. Amplified PCR products were separated using a 1% (w/v) agarose gel and visualized using ethidium bromide. Samples presenting bands around 300 bp in length were selected for TOPO cloning using fresh PCR products. A *mcyA* clone library was generated from the amplified PCR products by insertion into pCR4-TOPO TA vector (TOPO TA cloning kit Invitrogen/Life Technologies, Burlington, ON, Canada) and transformed into chemically competent One Shot TOP10 *Escherichia coli* cells. DNA sequencing was performed (Génome Québec Innovation Centre, Montréal, QC, Canada) and the resulting sequences were trimmed and dereplicated. Sequence alignment and phylogeny was completed using Mega 7.0 [65]. For a succinct comparison with previous studies, *mcyA* sequences generated in this study were clustered at 99% identity using UCLUST [66]; the most abundant sequence in the cluster for each toxin-producing species was then used as the reference sequence for phylogenetic comparison. To compare the reference sequences from this study with *mcyA* sequences from known toxin producing species, a Maximum-likelihood tree was generated.

### 5.5. Enzyme Linked ImmunoSorbent Assay (ELISA)

ELISA was used to analyze total microcystins when chl-a exceeded 5 µg/L. This assay (PN 520011OH, Abraxis LLC; Warminster, PA, USA) quantifies the ADDA moiety present in all microcystins and is expressed in terms of MC-LR equivalents. Filter-retained particulate microcystins (100–750 mL, depending on site) were extracted in 10 mL of Milli-Q water using ultrasound (three 60 s bursts with a 60 s pause between bursts) and centrifuged to pellet the cellular and filter debris. Extracts were then filtered through a 25 mm GF/F filter for the removal of suspended debris and MC concentrations were measured following the method described by Fischer et al. [67]. MC concentrations were back calculated to account for the concentration step. The Abraxis kit has a detection limit of 0.1 µg/L.

### 5.6. Mass Spectrometry Multiple Reaction Monitoring (MS-MRM)

Mass spectrometry multiple reaction monitoring (MS-MRM) was used to analyze samples from all collection sites for multiple MC congeners as well as STX and CYN (Table 5). Sample filters (collected as described above) were sonicated for 30 s in 2 mL of 80:20:0.1 acetonitrile: water: formic acid and extracted for four hours at 4 °C [68]. The extracts were filtered through 0.2 or 0.45 µm syringe filters into autosampler vials for high performance liquid chromatography tandem mass spectrometry (HPLC-MS/MS) analysis. The toxin extracts were analyzed on an Agilent 1200 series HPLC in-line with an Agilent 6410 triple quadrupole mass spectrometer (Agilent, Stanta Clara, CA, USA) fitted with an electrospray ionization source. The autosampler was maintained at 8 °C and injected 10 µL (quantitative analysis) or 40 µL (qualitative analysis) of sample. The analytes were passed through a column shield prefilter (MAC-MOD Analytical, Inc., Chadds Ford, PA, USA) and loaded onto a Luna C18(2), 3-µm particle size, 150 × 3 mm column (Phenomenex Corporation, Torrance, CA, USA) heated to 35 °C with 100% mobile phase A (90% water, 10% acetonitrile, 0.1% formic acid) at a flow rate of 0.4 mL/min. Initial conditions were maintained for two minutes, and analytes were eluted over a six-minute gradient from 0–90% mobile phase B (100% acetonitrile, 0.1% formic acid) followed by three minutes at 90% mobile phase B, before returning to initial conditions for three minutes. MS/MS analysis was carried out using Agilent MassHunter Data Acquisition software (version B.02.01, Agilent, Santa Clara, CA, USA). Samples were run in positive ion mode by MS-MRM (see Table 5 for transitions) and full scan mode (*m*/*z* 100–1200). Data were analyzed using Agilent MassHunter Qualitative Analysis software (version B.03.01, Agilent, Santa Clara, CA, USA). A standard curve (1/y^2^ weighting) was established for each toxin (except MC-LW, which was quantified using the MC-LR standard curve) by integrating the peak area of the quantifier ion from duplicate standards (6 concentrations ranging from 0–10 ng/µL), with a limit of detection of 0.5 ng on the Phenomenex column. Standards were prepared in methanol and analyzed in the same manner as the samples. To measure the amount of each toxin in the samples, the peak area of the quantifier ion was compared to the appropriate standard curve (Appendix A). The limit of detection of each toxin in water is 0.0003–0.0009 μg/L Microcystin (varies based on congener), 0.0005 μg/L Cylindrospermopsin, and 0.0009–0.0013 μg/L Saxitoxin. Standards for toxin analysis included various sources for microcystins including Enzo Life Sciences (Farmingdale, NY, USA), Cayman Chemical (Ann Arbor, MI, USA), Greenwater Laboratories (Palatka, FL, USA), and CCS purification. Pure saxitoxin standards were purchased from Cayman Chemical (Ann Arbor, MI, USA) and additional material was isolated from a toxic strain of Anabaena circinalis (obtained from Dr. Brett Neilan). Cylindrospermopsin standards were obtained from Dr. Brett Neilan.

### 5.7. Statistical Methodolog

Statistical relationships between nutrient concentrations, toxin concentrations, and genetic analysis was determined by calculating Pearson’s correlation coefficient using R version 3.5.3.

## Figures and Tables

**Figure 1 toxins-11-00587-f001:**
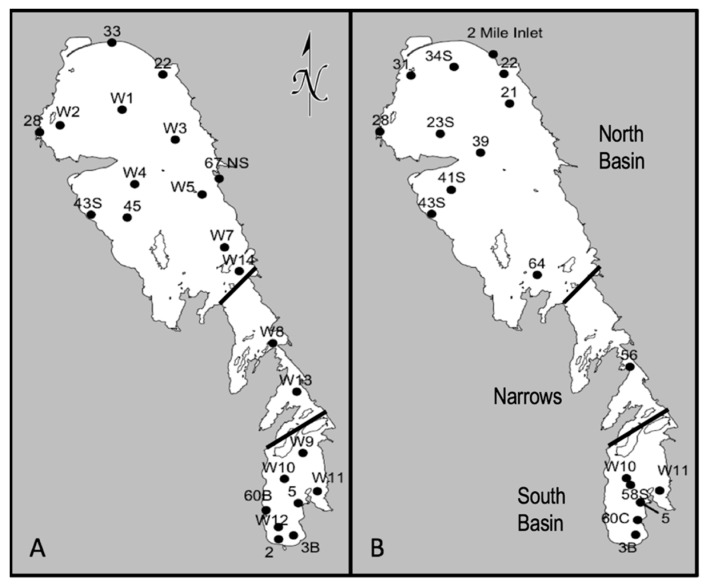
Lake Winnipeg sampling locations. (**A**) 23 sites were sampled in June for the summer 2013 season and (**B**) 18 sites were sampled in September for the fall 2013 season.

**Figure 2 toxins-11-00587-f002:**
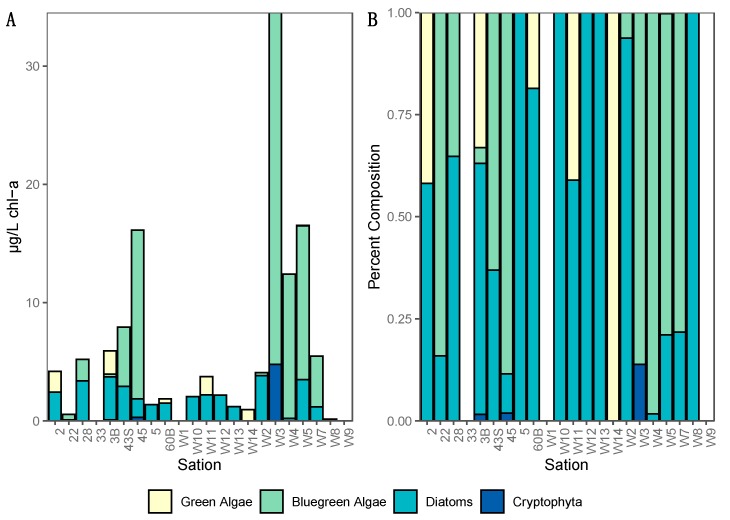
Total chlorophyll-a concentration (**A**) and % phylum composition (**B**) from flouroprobe analysis at summer sites in Lake Winnipeg. Sampling depths for flouroprobe data was approximately 1 m from the water surface.

**Figure 3 toxins-11-00587-f003:**
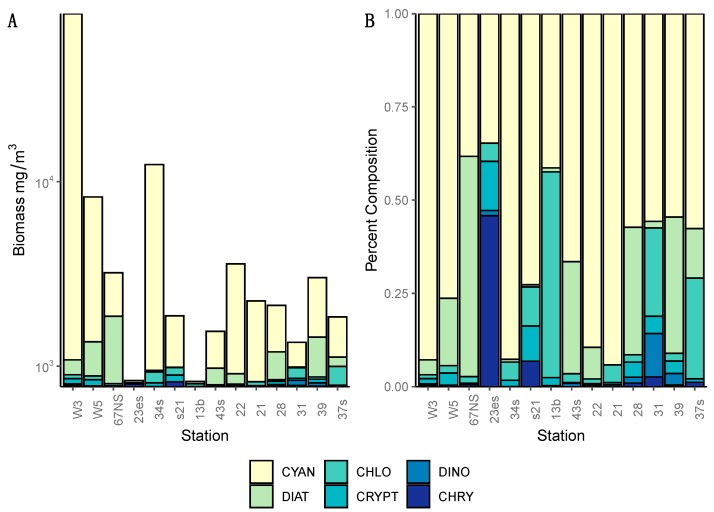
Total biomass (**A**) and % phylum composition (**B**) for eight summer sites and six fall sites in Lake Winnipeg. The dominant taxa in 11 of the 14 sites was cyanobacteria, consisting primarily of *Aphanizomenon flos-aquae* complex and *Dolichospermum* spp. The eight summer sites were W3, W5, 67NS, 23es, 34s, s21, 13b, and 43a. The six fall sites were 22, 21, 28, 31, 39, and 37s. Abbreviations for taxa are cyanobacteria (CYAN), diatoms (DIAT), chlorophytes (CHLO), cryptophytes (CRYPT), dinoflagellates (DINO), and chrysophytes (CHRY).

**Figure 4 toxins-11-00587-f004:**
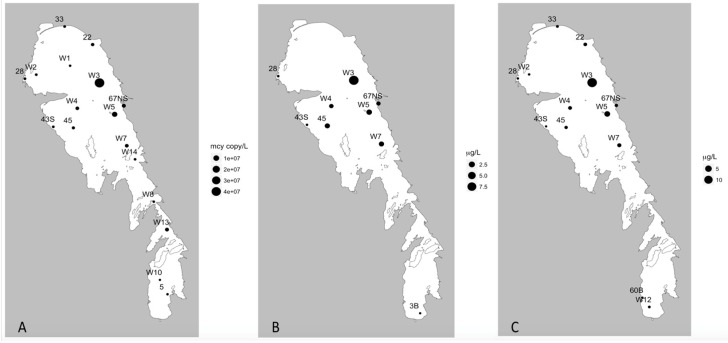
Relation of *mcyE* gene presence and actual microcystins (MC) toxin concentrations from the summer 2013 season. (**A**) *mcyE* gene concentration (gene copy/L), (**B**) microcystin concentration via ELISA assay (μg/L), and (**C**) microcystin concentration via Mass Spectrometry Multiple Reaction Monitoring (MC-MRM) analysis (μg/L).

**Figure 5 toxins-11-00587-f005:**
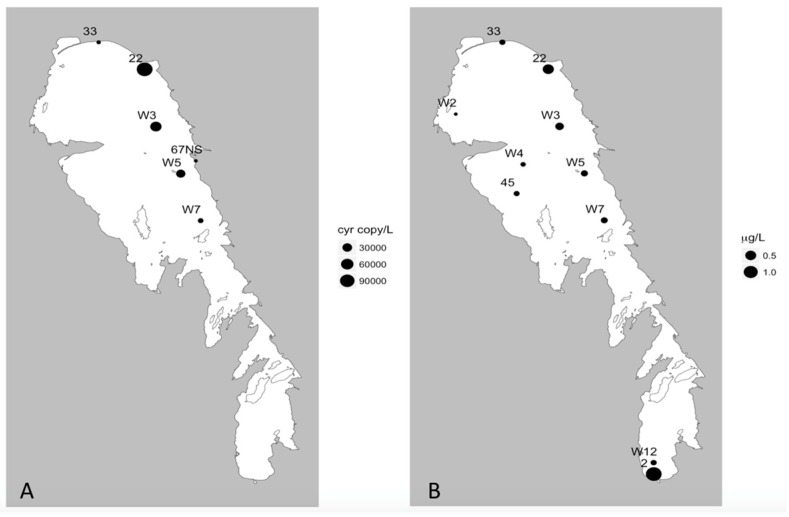
Presence of *cyrA* genes and cylindrospermopsin concentrations from the summer 2013 season. (**A**) *CyrA* gene concentration (gene copy/L) and (**B**) cylindrospermopsin concentration via MC-MRM analysis (μg/L).

**Figure 6 toxins-11-00587-f006:**
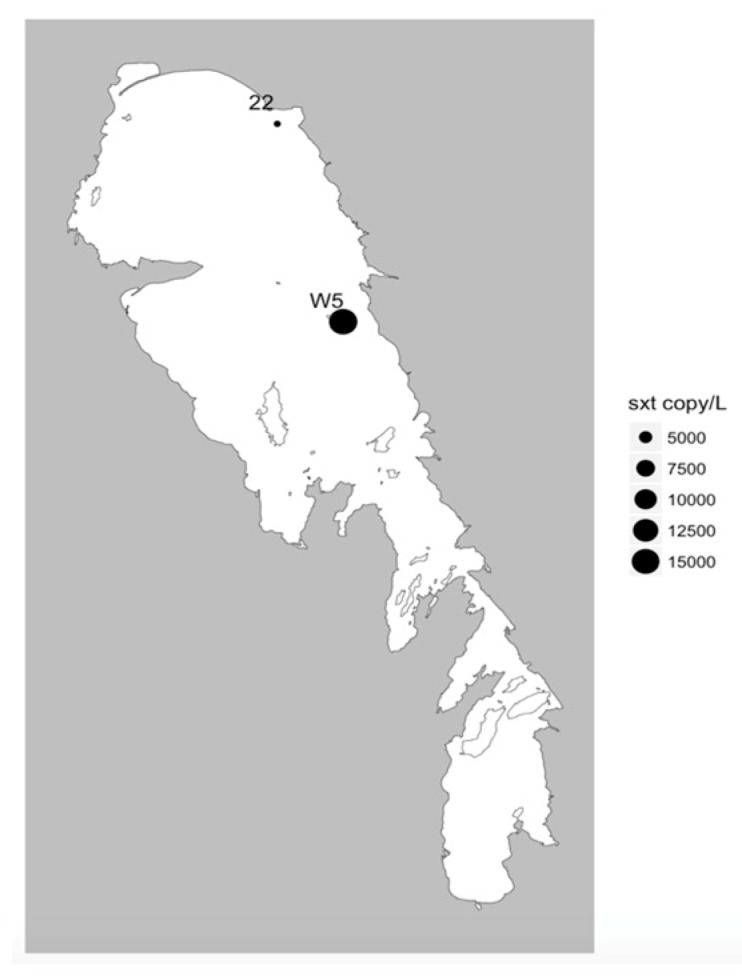
*SxtA* gene concentration (μg/L) from the summer 2013 season.

**Figure 7 toxins-11-00587-f007:**
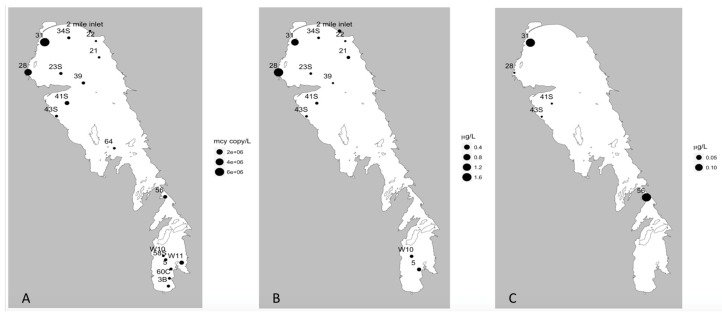
Relation of *mcyE* gene presence and actual MC toxin concentrations from the fall 2013 season. (**A**) *mcyE* gene concentration (gene copy/L), (**B**) microcystin concentration via ELISA assay (μg/L), and (**C**) microcystin concentration via MC-MRM analysis (μg/L).

**Figure 8 toxins-11-00587-f008:**
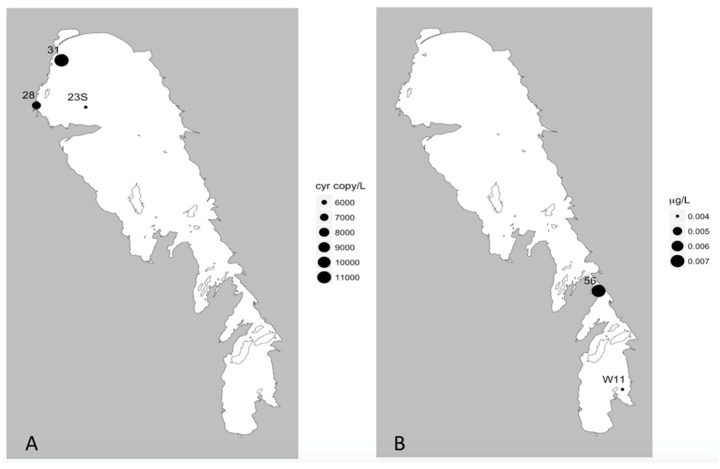
Presence of *cryA* genes and cylindrospermopsin concentrations from the fall 2013 season. (**A**) *CyrA* gene concentration (gene copy/L) and (**B**) cylindrospermopsin concentration via MC-MRM analysis (μg/L).

**Figure 9 toxins-11-00587-f009:**
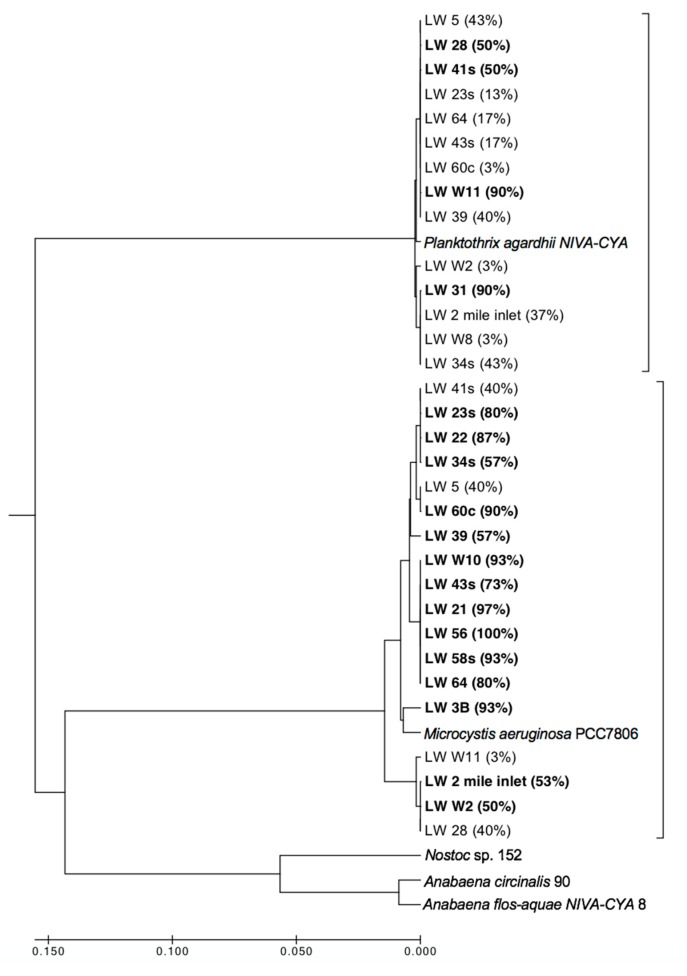
Phylogenetic tree of potential MC producers from 18 sites of the fall 2013 season and two sites from the summer 2013 season. All *mcyA* genes sequenced map to either *Planktothrix agardhii* or *Microcystis aeruginosa*. Bolded sites represent the dominant species (<50%) at that site.

**Table 1 toxins-11-00587-t001:** Dominant organisms at several sites in both summer and fall 2013 as determined by microscopic counts and given in percent total biomass and percent cyanobacterial biomass. Cyanobacterial species are highlighted in grey.

Station	Basin	Date	Taxa	% Total Biomass	%C Cyanobacteria Biomass
**SUMMER**					
W3	N	26-Jul-13	*Dolichospermum spp.*	68	74
			*Aphanizomenon flos aquae complex*	19	20
W5	NE	26-Jul-13	*Dolichospermum spp.*	54	72
			*Aphanizomenon flos aquae complex*	21	27.5
			*Aulacoseira ambigua*	6	
67NS	N	27-Jul-13	*Aphanizomenon flos aquae complex*	37	74
			*Dolichospermum spp.*	12	36
			*Aulacoseira ambigua*	51	
23es	N	27-Jul-13	*picocyanobacteria*	29	86
			large chrysophytes *(Ochromonads)*	25	
			*Rhodomonas minuta*	10	
34s	N	28-Jul-13	*Dolichospermum spp.*	78	84
			*Aphanizomenon flos aquae complex*	12	13
s21	N	28-Jul-13	*Dolichospermum spp.*	57	79
			*Aphanizomenon flos aquae complex*	9	12
			*Ochromonas spp.*	12	
			*Botryococcus braunii*	10	
13b	N	29-Jul-13	picocyanobacteria	40	95
			picoeukaryotes	53	
43s	NW	31-Jul-13	*Dolichospermum spp.*	62	91
			*Stephanodiscus niagarae*	21	
			*Fragilaria crotonensis*	8	
**FALL**					
22	NE	22-Sep-13	*Aphanizomenon flos aquae complex*	86	96
21	N	25-Sep-13	*Aphanizomenon spp.*	90	96
28	NW	25-Sep-13	*Aphanizomenon flos aquae complex*	36	63
			*Planktothrix agardhii*	13	22
			*Stephanodiscus niagarae*	27	
31	NW	25-Sep-13	*Dolichospermum spp.*	18	47
			*Planktothrix agardhii*	8	22
			*Aulacoseira ambigua*	15	
			*Eudorina elegans*	14	
39	N	28-Sep-13	*Aphanizomenon flos aquae complex*	52	96
			*Stephanodiscus niagarae*	26	
37s	S	3-Oct-13	picocyanobacteria	36	63
			*Aphanizomenon flos aquae complex*	12	20
			small centrics	12	

**Table 2 toxins-11-00587-t002:** Summary of toxin quantification data for Summer 2013.

Toxin Quantification, Summer 2013										
Sampling Site	Lake Region	Gene Copy/L			MC MRM (µg/L)	MC ELISA (µg/L)	CYN MRM (µg/L)	SXT MRM (µg/L)	Water Temp (°C)	Water Depth (m)
		*Cyr*	*Sxt*	*Mcy*						
W9	S	0	0	0	0	nt	0	0	21.9	11.2
W13	NA	0	0	2,220,000	0	nt	0	0	22.2	9.1
W8	NA	0	0	70,000	0	nt	0	0	21.0	13.1
W14	NA	0	0	130,000	0	nt	0	0	20.6	10.1
W7	NE	6300	0	1,740,000	0.90	1.54	0.11	0	18.6	15.5
W5	NE	26,500	15,700	7,730,000	3.32	1.91	0.12	0	18.3	14.3
67NS	NE	3900	0	2,460,000	0.37	0.79	0	0	15.9	4.0
W3	NE	46,700	0	44,090,000	14.18	9.21	0.23	0	18.3	17.4
W1	NM	0	0	70,000	0.003	nt	0	0	18.7	23.5
22	NE	112,900	4000	1,050,000	0.53	nt	0.53	0	18.2	14.6
33	NE	4200	0	170,000	0.10	nt	0.07	0	17.6	9.1
W2	NW	0	0	200,000	0.02	nt	0.02	0	19.1	14.3
28	NW	0	0	240,000	0.02	0.09	0	0	19.9	8.8
W4	NW	0	0	1,700,000	0.64	0.85	0.04	0	18.5	16.5
43S	NW	0	0	160,000	0.01	0.08	0	0	18.3	7.6
45	NW	0	0	660,000	0.34	1.49	0.06	0	18.0	13.4
W10	S	0	0	8000	0	nt	0	0	20.8	11.0
2	S	0	0	0	0	nt	1.37	0	20.3	6.1
3B	S	0	0	0	0	0.07	0	0	20.6	6.1
5	S	0	0	40,000	0	nt	0	0	20.5	10.7
W12	S	0	0	0	0.07	nt	0.07	0	19.9	8.5
60B	S	0	0	0	0.02	nt	0	0	20.0	8.2
W11	S	0	0	0	0	nt	0	0	19.8	8.2

NW = north west basin; NM = north middle basin; NE = north east basin; NA = narrows; S = south basin. nt = not tested.

**Table 3 toxins-11-00587-t003:** Correlation coefficient chart for Lake Winnipeg summer and fall 2013 toxin analyses.

Season	qPCR-MRM	qPCR-ELISA	MRM-ELISA
Summer	1.0	0.99	0.99
Fall	0.63	0.78	0.50

**Table 4 toxins-11-00587-t004:** Summary of toxin quantification data for Fall 2013.

Toxin Quantification, Fall 2013										
Sampling Site	Lake Region	Gene Copy/L			MC MRM (µg/L)	MC ELISA (µg/L)	CYN MRM (µg/L)	SXT MRM (µg/L)	Water Temp (°C)	Water Depth (m)
		*Cyr*	*Sxt*	*Mcy*						
28	NW	7200	0	3,350,000	0.022	1.68	0	0	15.6	7.3
23S	NW	5800	0	230,000	0	0.02	0	0	16.1	15.8
31	NW	11,300	0	6,410,000	0.148	0.92	0	0	15.8	10.98
34S	NM	0	0	210,000	0	0.02	0	0	15.91	15
2 mile inlet	NE	0	0	100,000	0	0.06	0	0	12.5	2.7
22	NE	0	0	100,000	0	0.01	0	0	15.6	14.63
21	NE	0	0	110,000	0	0.07	0	0	15.8	16.15
39	NM	0	0	260,000	0	0.01	0	0	16	17.1
41S	NW	0	0	780,000	0.022	0.04	0	0	15.55	11.58
43S	NW	0	0	190,000	0.022	0.02	0	0	15.34	7.6
64	NW	0	0	140,000	0	nt	0	0	15.6	15.2
56	NA	0	0	480,000	0.154	nt	0.007	0	15.9	21.3
58S	S	0	0	230,000	0	nt	0	0	15.7	8.2
W10	S	0	0	120,000	0	0.07	0	0	15.5	10.7
W11	S	0	0	770,000	0	nt	0.004	0	15.1	8.2
5	S	0	0	210,000	0	0.14	0	0	15.5	11
60C	S	0	0	170,000	0	nt	0	0	14.6	8.8
3B	S	0	0	230,000	0	nt	0	0	14.1	6.1

NW = north west basin, NM = north middle basin, NE = north east basin, NA = narrows, S = south basin, nt = not tested.

**Table 5 toxins-11-00587-t005:** Multiple reaction monitoring transitions.

Toxin	Precursor Ion	Quantifier Ion
dcSTX ^1^	257.1	138.2
SXT ^1^	300	204
CYN ^2^	416.1	194.1
MC-([M+2H]2+) ^3^	512.8	135.1
MC-RR ([M+2H]2+) ^3^	519.8	135.1
MC-WR ([M+2H]2+) ^3^	534.8	135.1
MC-LR ^3^	995.6	135.1
MC-LY ^3^	1002.5	135.1
MC-YR ^3^	1045.5	135.1

Transitions obtained from ^1^ Dell’Aversano et al. [68,69], ^2^ Guzman-Guillen et al. [70], ^3^ Chorus and Bartram [71]. Abbreviations: CYN—cylindrospermopsin; MC—microcystin; STX—saxitoxin.

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
