# Peer review of "A Multiplex Analysis of Potentially Toxic Cyanobacteria in Lake Winnipeg during the 2013 Bloom Season"

_toxins, 2019, doi:10.3390/toxins11100587_

Round 1
Reviewer 1 Report
This manuscript reported the occurrence of harmful cyanobacteria and cyanotoxin in Lake Winnipeg. The data presented here can be, to some degree, useful to aid the local water management in terms of understanding the situation of harmful cyanobacterial blooms and cyanotoxins. For the reviewer, the following points need to be addressed before publication. 1. The title of the present manuscript can be improved. In this work, the authors took the lake samples in summer and fall of 2013. However, only a small proportion of sampling sites were covered in two times of sampling schedule. Therefore, the data regarding to the seasonal variation should be limited to several sites, not at a large scale. 2. The authors should discuss the “weird” difference in MC detection between by HPLC-MS/MS/MS, and by the ELISA. Generally, the ELISA should detect more microcystin values than the chromatography-based methods. However, in Table 5 and 7, MRM-MS even detected much more microcystins than ELISA. This need to be addressed. 3. Table4 reflected the data about the composition of cyanobacteria. The current format is not easy to follow. Therefore, a figure format might be much better if possible. 4. The genetic data indicates the existence of Microcystis-like mcyE in the lake samples. However, the compositional data did not show the significant proportion of Microcystis. Why? 5. Figure 8, the Latin name of cyanobacteria species should be italicized.Author Response
The title of the present manuscript can be improved. In this work, the authors took the lake samples in summer and fall of 2013. However, only a small proportion of sampling sites were covered in two times of sampling schedule. Therefore, the data regarding to the seasonal variation should be limited to several sites, not at a large scale.
The title was changed to remove the seasonal variation and to address the date at which the data was obtained.
The authors should discuss the “weird” difference in MC detection between by HPLC-MS/MS/MS, and by the ELISA. Generally, the ELISA should detect more microcystin values than the chromatography-based methods. However, in Table 5 and 7, MRM-MS even detected much more microcystins than ELISA. This need to be addressed.
For the most part, our data does follow the trend where ELISA will report higher values than MRM. For the low samples that do not follow this trend, ELISA has lower sensitivity than HPLC methods. Therefore at low concentrations more can be detected using MRM procedures. For the higher sample (14 vs. 9), one possible explanation is human error with the dilution that had to take place to get the sample within the standard curve. Unfortunately, these samples are no longer available to re-test.
Table4 reflected the data about the composition of cyanobacteria. The current format is not easy to follow. Therefore, a figure format might be much better if possible.
Table was expanded on to make it more clear. Cyanobacteria species are still highlighted, but each species is in its own row with columns for % total biomass and % cyanobacteria biomass.
The genetic data indicates the existence of Microcystis-like mcyA in the lake samples. However, the compositional data did not show the significant proportion of Microcystis. Why?
Because we were just looking for the mcyA gene when sequencing, it is possible that the hits from each site only represent a few cells, and they were not dominant bloom forming organisms in 2013 (therefore, were not in a high enough proportion to be found through manual cell counts). The state of the Lake report for Lake Winnipeg addresses the fact that Microcystis blooms can occur in Lake Winnipeg, but they are not consistent. This information was added to the discussion section.
Figure 8, the Latin name of cyanobacteria species should be italicized.
Figure 8 has been corrected to have the species italicized.
Reviewer 2 Report
The manuscript presents quite ‘old’ data (from 2013) on potentially-toxic cyanobacteria from Lake Winnipeg.
This manuscript requires clarification of several aspects before acceptance:
Line 33 – the three zones were not highlighted in Fig. 1. As Fig. 1 is missing the North-south orientation, it makes difficult to guess these zones.
Fig. 1 – why the summer and fall sampling locations were so different, with a few exceptions? It was not explained in the text.
Fig. 2 - Summer and fall data were presented mixed together (these were not ordered separately). Not a good presentation. This order does not follow the order used in Table 4.
Fig. 2 – these colours are not understandable for readers who can only print in black/white.
Line 265 – this is not correct as STX congeners were not found (so, only 2 – not 3- toxins were found). The expression ‘cyanotoxins’ is not the most adequate, as these abbreviations refer to toxin groups, not individual toxins.
Line 352 – What is the meaning of ‘2’ in the sentence «are settled in a 2 Utermöhl settling…»
Section 5.6. The standard’s sources used for the LC-MRM analysis were not detailed.
Line 457 – I find it unusual that all toxins presented the same LOD.
Table 8 – why only for saxitoxins the authors presented a reference in the Table legend?
Author Response
The manuscript presents quite ‘old’ data (from 2013) on potentially-toxic cyanobacteria from Lake Winnipeg.
The title has been changed to address this fact, as well as parts of the discussion.
Line 33 – the three zones were not highlighted in Fig. 1. As Fig. 1 is missing the North-south orientation, it makes difficult to guess these zones.
A compass rose was added to figure 1 to aid the reader in distinguishing the regions. The regions have been split up and labeled.
Fig. 1 – why the summer and fall sampling locations were so different, with a few exceptions? It was not explained in the text.
The stations differed based on the needs of several research cruises, and the ships went to different regions based on the needs of all researchers. This information was added in the methods.
Fig. 2 - Summer and fall data were presented mixed together (these were not ordered separately). Not a good presentation. This order does not follow the order used in Table 4.
The figure was fixed to follow the same order as in Table 4, separating the summer and fall sites.
Fig. 2 – these colours are not understandable for readers who can only print in black/white.
Colorbrewer was used to find a color palette that would be both colorblind and printer friendly.
Line 265 – this is not correct as STX congeners were not found (so, only 2 – not 3- toxins were found). The expression ‘cyanotoxins’ is not the most adequate, as these abbreviations refer to toxin groups, not individual toxins.
This sentence was corrected to state that two toxin groups (MC and CYN) were detected via ELISA and MC-MRM, and the genetic capability to produce all three toxin groups was detected via qPCR.
Line 352 – What is the meaning of ‘2’ in the sentence «are settled in a 2 Utermöhl settling…»
The 2 was a typo, and has been removed.
Section 5.6. The standard’s sources used for the LC-MRM analysis were not detailed.
Standard sources were added to the methods section.
Line 457 – I find it unusual that all toxins presented the same LOD.
They do not have the same LOD. LOD for each toxin was only included in a reference to the EPA method 525. The methods section on LC-MRM has been expanded to include toxin specific limits of detection based on this method for clarification.
Table 8 – why only for saxitoxins the authors presented a reference in the Table legend?
The references for the other toxin types were added to the table.
Reviewer 3 Report
This manuscript reports the seasonal variation of cyanobacterial toxins and the genes that are related to production of these cyanobacterial toxins (microcystins, cylindrospermopsin, saxitoxin)(cHAB) in Lake Winnipeg, Canada, the world’s 12 largest lake. The authors collected surface water samples from 23 sites during summer and 18 sites in fall in 2013. They used qPCR, ELISA, and LCMSMS to measure cHAB and its related genes. This study is important to define the baseline composition of this lake cHAB to measure potential changes due to dreissenid colonization as they mention. Data is solid and valuable. This paper is recommended to be published after minor revision.
Some of the fonts of scale (X and Y axis) of the graphs in Figure 2 are too small. They should be enlarged to be shown clearly.
The font in Table 6 is also too small.
line 433; please show typical LCMS chromatograms of sample and standard in the text.Author Response
Some of the fonts of scale (X and Y axis) of the graphs in Figure 2 are too small. They should be enlarged to be shown clearly.
Fonts for figure 2 were increased to make it more readable.
The font in Table 6 is also too small.
Table 6 was replaced with an editable table that removed the extraneous data. The font size can now be adjusted per editors requirements
line 433; please show typical LCMS chromatograms of sample and standard in the text.
Examples for MC and CYN have been included as part of the supplementary materials.
Reviewer 4 Report
The paper reports the chemical parameters of water, phytoplankton composition, cyanobacteria dominant species and toxin profile (analytically and molecularlly) in Lake Winnipeg. The ms is generally OK but:
The data is from 2013 and are of archival value. Is there any newer data (e.g. nutrient profiles) available to discuss? The data is rather old and limited, and this should be acknowledged. The statistical methodology is not reported. Some minor issues are highlighted directly on pdf file
Author Response
The data is from 2013 and are of archival value. Is there any newer data (e.g. nutrient profiles) available to discuss?The data is rather old and limited, and this should be acknowledged.
No new data is available to discuss. The purpose of this paper was to print "old" data such that others can use it as a comparison for future work. To address this issue, the title has been changed to reflect that the data set is old, as well as a few sentences in the discussion.
The statistical methodology is not reported.
A sentence about correlation coefficient analysis was added to the methods section.
Some minor issues are highlighted directly on pdf file
All minor issues were corrected in the updated manuscript. Grammatical issues were corrected, and missing information was added in sections that needed additional information. All of the statistical notation was also corrected. BMAA information was added in the discussion to address another known cyanotoxin that was not covered in this paper. Sample number for analysis was added to the methods section.
Round 2
Reviewer 2 Report
I am pleased with the improvement's made to the manuscript.
Author Response
Thank you for your comments.